# Enzymatic Activity and Nutrient Profile Assessment of Three *Pleurotus* Species Under Pasteurized *Cenchrus fungigraminus* Cultivation

**DOI:** 10.3390/cimb47030143

**Published:** 2025-02-22

**Authors:** Nsanzinshuti Aimable, Hatungimana Mediatrice, Irambona Claude, Jules Biregeya, Yingping Hu, Hengyu Zhou, Penghu Liu, Jing Li, Zhanxi Lin, Guodong Lu, Dongmei Lin

**Affiliations:** 1National Engineering Center of Juncao Technology, College of Life Science, Fujian Agriculture and Forestry University, Fuzhou 350002, China; nsanziaima@gmail.com (N.A.); mediatunga@gmail.com (H.M.); claudebiofafu@gmail.com (I.C.); biregeyakayihura2020@gmail.com (J.B.); hyp111419@126.com (Y.H.); hengyou@fafu.edu.cn (H.Z.); phliu1982@163.com (P.L.); fafulijing@fafu.edu.cn (J.L.); lzxjuncao@163.com (Z.L.); 2Rwanda Agriculture and Animal Resources Development Board, Kigali P.O. Box 5016, Rwanda

**Keywords:** enzyme activity, *Cenchrus fungigraminus*, *Pleurotus* species, nutrients, biological efficiency

## Abstract

Oyster mushrooms are regarded as one of the most significant edible mushrooms in terms of commercial value because of their rich nutritional profile. Many bioactive extracts from *Pleurotus* species have been found to exhibit antitumor and antioxidant activities. However, to grow oyster mushrooms in this study, the pasteurized *Cenchrus fungigraminus* was used as culture material, a type of grass that proliferates and has a high root growth rate. It contains high levels of sugar and protein and yields a large amount of biomass. Because of these characteristics, it is considered an efficient and cost-effective energy crop with various applications, including phytoremediation and fodder production. A pasteurization technique for this grass that is suited for the simplest formulation is simple and cost-effective for growing oyster mushrooms on small farms. This study used pasteurized *Cenchrus fungigraminus* as a substrate to grow three mushroom species: *Pleurotus ostreatus*, *Pleurotus pulmonarius* and *Pleurotus florida*. The aim was to evaluate their enzyme activities, growth rate, and yields. The findings demonstrated that the average growth rate of three species grown in pasteurized *C. fungigraminus* was between 25 days and 36 days. Therefore, the mycelium growth rate of *P. ostreatus* was faster than other *pleurotus* species in this study. The highest biological efficiency was recorded with *P. ostreatus* at 78.23%, then *P. pulmonarius at* 59.88, and lastly, 39.66% *P. florida*. The changes in five enzyme activities in distinct developmental stages of three different *pleurotus* species were evaluated. Therefore, the laccase had the highest peak with 13.8 U/g on the 20th day during the growth phase and gradually decreased to the fruiting body stage of *P. ostreatus*. The expression of manganese peroxidase reached the highest activity of 3.6 U/g in *P. ostreatus* compared to *P. florida* and *P. pulmonarius* on the 10th day. The expression of other enzymes varied between species and developmental phases. The results indicate the usefulness of pasteurized *C. fungigraminus* for cultivating *Pleurotus* species and expression enzyme activity in different *Pleurotus* species.

## 1. Introduction

Mushrooms are macrofungi with distinct fruiting bodies, and edible mushrooms have been recognized as a food source by the Food and Agriculture Organization due to their nutritional quality. Mushrooms contain more protein than either fruits or vegetables [1]. Growing mushrooms can significantly enhance sustainable livelihoods for both urban and rural poor people, as they easily integrate with other income-generating activities while needing minimal physical and financial resources (FAO, 2004). *Pleurotus* species belong to the family of *Tricholomataceae* and are usually found clustering naturally on dead trees in the spring season. *Pleurotus* species are popular and widely cultivated worldwide, mostly in Asia, America, and Europe because of their simple, low-cost production technology and high biological efficiency [2].

*Pleurotus* species are efficient lignin degraders that can grow on various agricultural wastes at various temperatures and have economic, ecological, and medicinal values. Chand et al. proposed that growing mushrooms can serve as a viable method for increasing income in developing nations because it offers low production costs, significant profits, and rapid financial returns [3]. The substrate composition are the most important factors that affect the enzyme activity, quality, and yield of edible fungi [4]. Substrates containing lignocellulose are needed to grow mushrooms, which are the sources of carbon-nitrogen and other essentials [5].

*Cenchrus fungigraminus* is a perennial grass with high cellulose and crude protein content, capable of yielding up to 200 tons per hectare as biomass [6], and has abilities to grow at a wide range of temperatures utilizing various lignocelluloses, which is very necessary for mushroom cultivation. Moreover, they can colonize and degrade many lignocellulosic materials and other wastes produced in agricultural, forest, and food-processing industries [7]. The cultivation of edible mushrooms might be the only current process that combines the production of protein-rich food with the reduction in environmental pollution [8].

Cultivating oyster mushrooms involves a few crucial steps, one of which is the pasteurization or sterilization of the substrate. Pasteurization of substrate is a simple, cost-effective, and efficient method for treating organic materials used in mushroom cultivation [9]. It reduces the harmful pathogens and pests that could compete with or damage mushroom cultures [10]. Mushrooms produce many extracellular enzymes that can degrade lignocellulose materials into soluble compounds of a relatively low molecular mass. Those soluble compounds have remarkable nutritional value, and white rot fungi are the most ligninolytic microorganisms in nature [11]. They can decompose lignin-producing extracellular enzymes by a unique mechanism that has the market capacity to generate oxidative radicals, including hemicelluloses, cellulose, and lignin.

The enzyme system is nonspecific and it can decompose a wide variety of structurally different compounds [12]. Decomposition is performed by a complex set of enzymes, containing lignin peroxidase, laccase, and xylanase, involving microbes for the solubilization of insoluble macromolecules like cellulose, hemicelluloses, and lignin, based on the secretions of the enzymes in their substrate [13]. The fungi stand out for their adeptness in decomposing lignin, achieved through the external release of enzymes collectively referred to as ligninases, as depicted encompassing laccases, and manganese peroxidases [14]. Many fungi, bacteria, and others use carboxymethylcellulose as a food source. Those organisms produce cellulase, enhancing the enzyme system to aid in the breakdown of cellulose and its biological conversion to glucose which can utilized as energy [15]. These enzymes play pivotal roles in various fungal biological processes, underscoring their significance. Mushrooms are unique biota that contain degrading enzymes, allowing them to decompose complex food materials to obtain their necessary nutrients. They convert these nutrients into simpler compounds which they absorb and incorporate into their tissues [16].

Currently, there is limited research on enzyme expression levels in various developmental stages of mycelial growth in oyster mushrooms and also the use of pasteurized *Cenchrus fungigraminus* as a substrate. This study investigates the expression levels of five different enzyme activities during various developmental stages of mycelial growth in three species of oyster mushrooms and biocomponents.

## 2. Materials and Methods

### 2.1. Preparation of Primary and Secondary Spawn

This study was carried out at the Academy of Juncao Science and Technology, Fujian Agricultural and Forestry University. Three *Pleurotus* strains, specifically P969, P377, and P359, were initially sourced from various geographic areas in China [17] and gathered at Fujian Agricultural and Forestry University in Fuzhou City as illustrated in Table 1. These strains were preserved on Potato Dextrose Agar (PDA) medium for primary spawn, and second spawn were prepared in polyethylene bags filled with 500 g of sterile substrate, comprising 40% *Miscanthus floridulus*, 38% *Dicranopteris dicholoma*, 20% wheat bran, and 2% gypsum.

### 2.2. Preparation of Culture Medium and Cultivation

Fully grown plants were collected, reduced to a particle size ranging from 0.1 to 0.5 cm, and dried in the sun for three days. Once fully dried, the pasteurized *C. fungigraminus* grasses were soaked in regular water with the addition of 2% lime for 2 h. After two hours of soaking, the grass was removed from the limed water and thoroughly pressed to remove excess water. Following this, 20% wheat bran and an additional 2% lime were incorporated based on the quantity of *C. fungigraminus*. The bags were inoculated with 30 g of the mushroom second spawn of three different strains according to the Claude et al. (2024) method. Each polyethylene bag containing 500 g of substrates was sealed with perforated caps to allow for some aeration. Eighteen bags were prepared for each of the mushroom species. The inoculated bags were placed at room temperature to allow for incubation. To assess the growth rates among *Pleurotus* species, the duration needed for the substrate to colonize completely and for the initiation of primordial structures was documented for each bag. Once the bags had fully colonized, they were opened and moved to a growing room maintained at 25 °C and 85% humidity [18]. Humidity levels were kept consistent through regular water spraying. Mycelial samples were collected by randomly choosing one bag from each species. After thorough mixing, three replicate samples were taken during the substrate inoculation phase, mycelial colonization phase, and fruiting body phase. Each bag’s yield from two consecutive flushes was recorded to assess the harvest. Samples of both the fruiting bodies and the colonized substrate were immediately frozen with liquid nitrogen and stored at −80 °C for future use [19]. The biological efficiency (BE) for each species was then calculated by the following formula [20].$$BE=\frac{Fresh\;weight\;of\;fruit\;body\times 100}{The\;dry\;weight\;of\;the\;substrate}$$

### 2.3. Determination of Enzyme Activity

Enzyme activity assays for laccases, amylase, carboxymethylcellulose (CMCase), manganese peroxidase (MnP), and xylanases were conducted during the mycelial and fruiting phases of various *Pleurotus* strains (P969, P377, and P359) [21]. The assays were conducted using the Enzyme-Linked Immunosorbent Assay (ELISA) method with the conventional ELISA kits acquired from Shanghai Uplc-MS Testing Technology Co., Ltd.in Shanghai, China. Additionally, ABTS was utilized due to its recognized specificity and sensitivity in measuring enzyme activities [22].

The procedure evaluated laccase activity by centrifuging a 1 g sample with an extraction solution at 10,000× *g* for 10 min at 4 °C. For the control sample, 25 μL of boiled samples was combined with 25 μL of test samples in a 2 mL EP tube, followed by the addition of 150 μL of working fluid containing ABTS to the test sample. An equal volume of 150 μL of working fluid was added to the control sample as well. Both samples were then incubated in a water bath at 60 °C for 3 min. The oxidation of ABTS was monitored by measuring the increase in absorbance at 420 nm, recorded every 30 s over a total duration of 180 s, and the quantity of enzyme required to oxidize 1 nmol of substrate ABTS per gram of sample per minute is one unit of enzyme activity. To assess the total activity of alpha and beta amylase, a crude amylase solution was mixed with 4 mL of double-distilled water and shaken thoroughly. The resulting reducing sugars were quantified using a spectrophotometer/microplate reader and preheated for 40 min, with the wavelength set to 540 nm [23].

Carboxymethylcellulase was extracted by centrifuging one gram of the sample with 1 mL of extract at 8000 rpm for 10 min at 4 °C. The measurement of reducing sugars was conducted using the 3,5-dinitrosalicylic acid method. In each test tube, 50 μL of reagent one, 200 μL of reagent two, and 50 μL of distilled water were added, followed by 50 μL of the sample. To facilitate cooling, an additional 250 μL of distilled water was incorporated into all tubes. Finally, 200 μL of the resulting solution was transferred to a trace quartz cuvette for analysis [24].

In the MnP procedure, a sample weighing 0.1 g was measured, and 1 mL of reagent one was added. The mixture was then blended in an ice bath at 4 °C before being centrifuged at 10,000 rpm for 10 min. For the control sample, 120 μL of reagent one was pipetted. For the sample analysis, 100 μL of the supernatant was transferred into a 1.5 mL Eppendorf tube, followed by the addition of 20 μL of reagent two, 40 μL of reagent three, 20 μL of the sample, and 20 μL of reagent four into their respective test tubes. Each test tube was thoroughly mixed and incubated in a water bath at 30 °C for 10 min. Absorbance levels were recorded at 465 nm using a spectrophotometer [25].

Xylanase activity was assessed by preparing D-xylose standard solutions with concentrations ranging from 100 to 600 µg/mL. A mixture was created by combining 1 µL of D-xylose with 1.5 mL of DNS reagent, which was then boiled for 7 min and subsequently cooled. The absorbance of this mixture was measured at 550 nm [26].

### 2.4. Determination of Nutritional Content of Pleurotus Species

The determination of biocomponent content such as polysaccharides, fiber, carbohydrates, fat, proteins, amino acids, crude ash, and heavy metals such as cadmium, arsenic, lead, and mercury was performed in the fruiting body of *Pleurotus ostreatus*, *Pleurotus pulmonarius* and *Pleurotus florida* [27]. A sample weighing 0.05 g was placed in a test tube with 1 mL of water and then homogenized in a water bath at 100 °C for 2 h. After this, the mixture was centrifuged at 10,000 rpm for 10 min, and the supernatant was carefully collected. Next, 0.2 mL of the supernatant was combined with 0.8 mL of anhydrous ethanol. This mixture was then used to quantify polysaccharides at 490 nm according to the method used by Sawangwan et al. (2018) [28].

The determination of crude fiber was performed in accordance with GB/T 5009.10-2003. The crude ash content was determined according to GB/T12532-2008. The standard working solution was created by combining 100 volumes of BCA (Bicinchoninic Acid protein assay kit) reagent A with 2 volumes of BCA reagent B. Reagent A was prepared by dissolving 1 g of sodium bicinchoninate, 2 g of sodium carbonate, 0.16 g of sodium tartrate, 0.4 g of NaOH, and 0.95 g of sodium bicarbonate in 50 mL of distilled water [29]. This solution was then brought to 100 mL with distilled water, and the pH was adjusted to 11.25 using 10 M NaOH. Reagent B was made by dissolving 0.4 g of cupric sulfate (5 × hydrated) in 5 mL of distilled water and then adding distilled water to reach a total volume of 10 mL. The absorbance of the known standard was recorded at 562 nm. The fat content and amino acids in the fruiting bodies were examined using the RP-HPLC method outlined by Mustafa et al. (2022). The ion exchange technique was employed to examine the fruiting body’s levels of Cd, Pb, As, and Hg [30].

### 2.5. Statistical Analysis

Data from three oyster mushrooms were analyzed using a one-way ANOVA test in SPSS software version 27 to compare the means of multiple groups. A post hoc analysis was conducted using the Duncan test at the 5% significance level to identify significant differences between group means. Graphs were designed and statistically analyzed using GraphPad Prism 9.5 [31].

## 3. Results

### 3.1. Yield and Biological Efficiency

The yield is a crucial factor in mushroom cultivation. In this research, we compared the yields of three *Pleurotus strains* P969, P359, and P377 across two flushes, all cultivated on pasteurized *C. fungigraminus*. Our results indicated that *P. ostreatus* produced the highest yield at 164.5 g per bag. As shown in Table 2, there was a significant increase in the yield of *P. ostreatus* (*p* ≤ 0.05), followed by *P. pulmonarius* with 117.6 g per bag, while *P. florida* yielded comparatively less. We also assessed the biological efficiency (BE) throughout the entire growth period of these oyster mushrooms. Our findings revealed that the BE of P969 was significantly high (*p* ≤ 0.05), reaching 78.23%; this was succeeded by P359 at 59.88% and P377 at 39.66%, all cultivated on pasteurized *C. fungigraminus.*

Moreover, the biological efficiency among different oyster mushroom strains grown on the same substrate demonstrated significant variations (*p* ≤ 0.05). These findings are consistent with those of Ahmed et al. (2024), who noted that *P. ostreatus* achieved the most significant biological efficiency across various substrates. Notably, the differences observed in stipe lengths and pileus diameters were not correlated with their respective yield quantities [32].

### 3.2. Nutrient Composition and Heavy Metals in the Fruiting Bodies

The nutritional analysis of the three *Pleurotus* species (P969, P359, P377) grown in pasteurized *Cenchrus fungigraminus* encompassed the examination of protein, ash, fat, fiber content, and polysaccharides, which show notable differences in their nutritional profiles [33]. Upon evaluation, it was observed that the nutritional composition of the fruiting body of *P. ostreatus* had the most significant protein and fiber contents, whereas *P. pulmonarius* had the most significant polysaccharide and fat contents, and *P. florida* had the most significant ash content as demonstrated in Table 3; the observed variance in means was statistically significant at the 0.05 interval for the nutritional components under consideration.

The concentrations of heavy metals, specifically cadmium, arsenic, lead, and mercury, were analyzed in the fruiting bodies of three oyster mushroom varieties. Previous studies have indicated that edible fungi can accumulate varying levels of heavy metals according to the report by Dowlati et al. (2021) [34]. As shown in Table 4, *P. pulmonarius* exhibited the most significant cadmium content at 0.136 mg·kg^−1^ compared to the other two strains. In contrast, the most significant concentrations of lead and arsenic were found in *P. ostreatus,* with levels of 0.628 mg·kg^−1^ and 0.7 mg·kg^−1^, respectively. Additionally, *P. florida* displayed the highest mercury concentration at 0.086 mg·kg^−1^. The study assessed four heavy metals in the fruiting bodies of oyster mushrooms cultivated in pasteurized *C. fungigraminus* substrates. The results indicated no significant difference at the 0.05 significance level, and the measured concentrations complied with international food standards. These findings are crucial for ensuring food safety and regulatory compliance and understanding the environmental implications of mushroom cultivation based on the WHO/FAO Expert Committee on Food Additives (JECFA) [35].

### 3.3. The Change in Enzymes on Mycelia Growing in the Different Growing Stages of Three Oyster Mushrooms

The variation in laccase activity among three strains of oyster mushrooms (P969, P377, P359) cultivated on pasteurized *C. fungigraminus* substrate was examined. As depicted in Figure 1, there exists significant variation in laccase enzyme activity throughout the various developmental stages of these mushrooms. Twenty days post-inoculation, the laccase enzyme activity for P969 reached its highest peak at 13.8 U/g, surpassing that of P359. Subsequently, this activity rapidly decreased during the mycelium growth phase. Notably, significant differences in laccase enzyme activity were observed among the three *Pleurotus* strains during the same growing period. These results suggest that laccase activity plays a crucial role in influencing the yield of different oyster mushroom strains.

The alteration of amylase activity in three varieties of oyster mushrooms (P969, P377, P359) cultivated on pasteurized *C. fungigraminus* substrate was examined. As illustrated in Figure 2, there was considerable variation in amylase enzyme activity among the three types of oyster mushrooms throughout different growth stages. The emergence of distinct enzyme activities was noted ten days post-incubation. Notably, the amylase enzyme activity of *P. pulmonarius* reached its peak on the 20th day at 0.32 U/g during the primordia formation stage, followed by a gradual decline during the fruiting body phase. Furthermore, significant differences in amylase activity were observed among the various oyster strains during the same growth period, indicating that the amylase enzyme activity varied and was influenced by the different oyster strains.

The activity of carboxymethylcellulase (CMCase) varied during the mycelium growth stage and gradually declined as the fruiting body developed in all *Pleurotus* species. As illustrated in Figure 3, ten days post-incubation, the CMCase activity in the *P. pulmonarius* species reached a peak of 1.4 U/g, surpassing that of both *P. ostreatus* and *P. florida*. This activity slowly decreased during the formation of the fruiting body. The elevated enzyme activity significantly impacts production. Notably, the carboxymethylcellulase activity observed in the growth phases of the three oyster mushroom species (P969, P377, P359) demonstrated significant differences throughout the entire growing process (*p* ≤ 0.05).

The change in manganese peroxidase activity in three oyster mushrooms (P969, P377, P359) cultivated on pasteurized *Cenchrus fungigraminus* substrate was studied. As illustrated in Figure 4, these three oyster mushrooms showed a clear decline in variations in MnP activity during all developmental stages. The appearance of change activities started ten days after inoculation. MnP activity in *P. ostreatus* showed the highest peak on the 10th day (3.6 U/g) in the primordia formation, then decreased gradually until the fruiting body stage. There was relatively low production of MnP in *P. pulmonarius,* which decreased progressively during the end of the life cycle. Therefore, the manganese peroxidase activity is most stable in the P969 strain, followed by P377, while P359 shows the least stability.

As depicted in Figure 5, various changes led to an increase in the xylanase enzyme activity of P969, P377, and P359, which were cultivated on pasteurized *C. fungigraminus* culture material throughout the growing period, from the mycelium growth stage to the primordial formation and finally to the mature stage. On the thirtieth day, the xylanase enzyme activity of P969 reached its peak at the fruiting body stage, measuring 1.1 U/g. Notably, significant differences in xylanase activity were observed among the three oyster strains during the same growth period. The results indicated that xylanase activity has a considerable impact on the production of the different oyster strains cultivated under identical culture conditions.

## 4. Discussion

The various oyster mushroom species grown on pasteurized *Cenchrus fungigraminus* substrate showed notable differences in biological efficiency (*p* ≤ 0.05). These results align with the findings of Ahmed et al. (2024), who indicated that *Pleurotus ostreatus* had the highest biological efficiency among various oyster strains and substrates [32]. This highlights the potential of *P. ostreatus* as an advantageous species for optimal substrate use and high-yield production, which is essential for commercial mushroom farming. Moreover, Margaret et al. (2023) found no relationship between pileus diameters and stipe lengths, including their respective counts and weights. This highlights the intricate interactions of morphological characteristics and underscores the necessity for thorough studies to grasp the growth dynamics of various mushroom species [36].

The protein contents recorded in this study for *P. ostreatus* (4.53 mg·g^−1^), *P. pulmonarius* (3.641 mg·g^−1^), and *P. florida* (2.99 mg·g^−1^) are very similar to the previously reported range of 3.0 to 4.0 mg·g^−1^ for *P. ostreatus* cultivated individually on six different sawdust substrates, as per Calabretti et al.’s (2021) findings. This notable rise in protein content emphasizes the nutritional enhancement achieved by the innovative application of pasteurized *C. fungigraminus* as a substrate [37]. The amounts of carbohydrates, amino acids, fats, and crude ash observed in this study also surpass the findings presented by Niyimbabazi et al. (2022). This indicates that the substrate not only facilitates high biological efficiency but also improves the nutritional quality of the mushrooms, enhancing their value as a food source [38].

The variations in heavy metal concentrations among different *Pleurotus* species are in agreement with the findings noted by Innes (2023). Furthermore, the heavy metal levels found in the fruiting bodies of the three analyzed species in this study fell within safe limits for vegetable consumption [39]. This guarantees that the produced mushrooms are safe to eat and adhere to health standards, which is crucial for consumer safety and market acceptance.

Significant differences in enzyme activity among the *Pleurotus* species were observed all developmental stages in this research. Moreover, *Pleurotus ostreatus* shows higher efficient enzyme activity compared to other species grown in *C. fungigraminus*. Research indicates that the conversion of lignocelluloses into soluble sugars by *Pleurotus* species is significantly influenced by various nonspecific oxidative enzymatic systems, primarily laccases and manganese peroxidases. Those findings were in alignment with previous work by Claude et al. (2024), which found that the growth of *Pleurotus ostreatus* on substrates such as *C. fungigraminus* had a notable impact on enzyme activity, although the activity varied at different developmental stages [18]. The substantial laccase activity seen in the three *Pleurotus* species P969, P377, and P359 aligns with previous research by Han et al. (2022), which recognizes it as the most effective and well-researched ligninolytic enzyme involved in lignin oxidation by white rot fungi. This heightened laccase activity is correlated with the *Pleurotus* species’ ability to colonize and decompose the resistant lignin polymer [40]. Earlier studies have indicated that enzyme expression is influenced by the type of substrate, the method of pasteurization used, and several external factors like temperature, pH, and specific fungal strains involved [41]. This study enhances our understanding of these dynamics and emphasizes the importance of enzyme activity in substrate breakdown and nutrient accessibility.

The importance of this research lies in its examination of enzyme activity within three *Pleurotus* species cultivated on pasteurized *Cenchrus fungigraminus* as a substrate, which plays a critical role in food security. This innovation could significantly enhance the sustainability and productivity of mushroom farming practices. Future studies should concentrate on detailed investigations into the kinetics and mechanisms of specific enzymes produced during the degradation process. Additionally, continual efforts to identify effective disinfection methods for pathogenic microorganisms affecting mushroom mycelium at various developmental stages will deepen the understanding of their functional characteristics. This strategy could maximize the potential in the realm of oyster mushroom cultivation, resulting in greater productivity and sustainability.

## Figures and Tables

**Figure 1 cimb-47-00143-f001:**
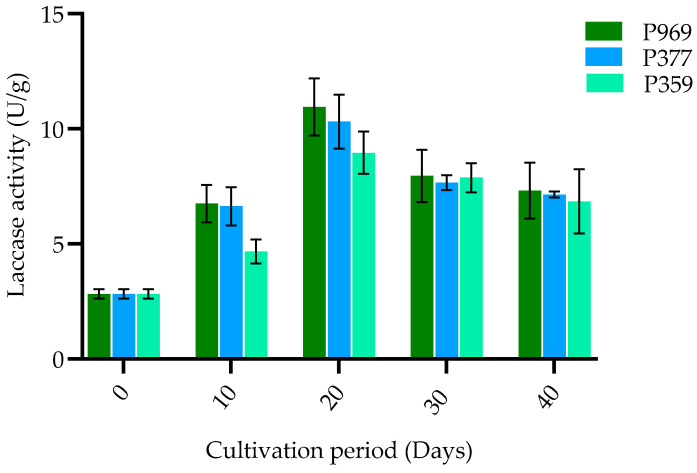
The bar plots show the change in laccase activity in P969, P377, and P359 cultivated on pasteurized *C. fungigraminus* substrate. The black bars in the plots indicate the standard error of the mean. U/g: units per gram of the sample; P969: *Pleurotus ostreatus*; P377: *Pleurotus florida*; P359: *Pleurotus pulmonarius*.

**Figure 2 cimb-47-00143-f002:**
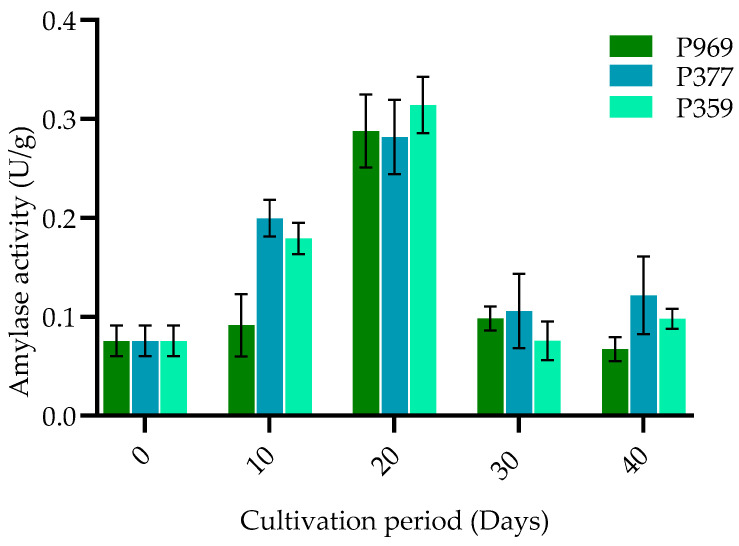
Bar plots showing the change in amylase activity in P969, P377, and P359 cultivated on pasteurized *C. fungigraminus* substrate. The black bars in the plots indicate the standard error of the mean. U/g: units per gram of the sample; P969: *Pleurotus ostreatus*; P377: *Pleurotus florida*; P359: *Pleurotus pulmonarius*.

**Figure 3 cimb-47-00143-f003:**
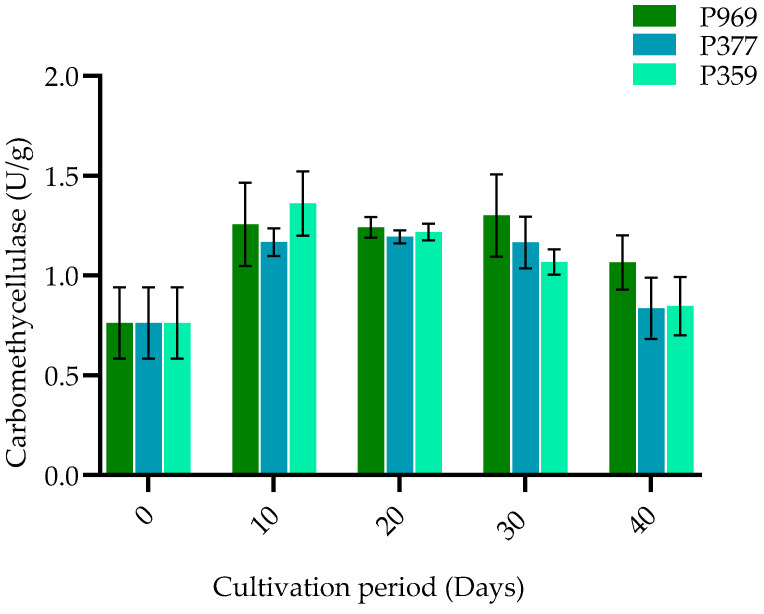
The bar plots show the change in carboxymethylcellulase activity in P969, P377, and P359 cultivated on pasteurized *C. fungigraminus* substrate. The black bars in the plots indicate the standard error of the mean. U/g: units per gram of the sample; P969: *Pleurotus ostreatus*; P377: *Pleurotus florida*; P359: *Pleurotus pulmonarius*.

**Figure 4 cimb-47-00143-f004:**
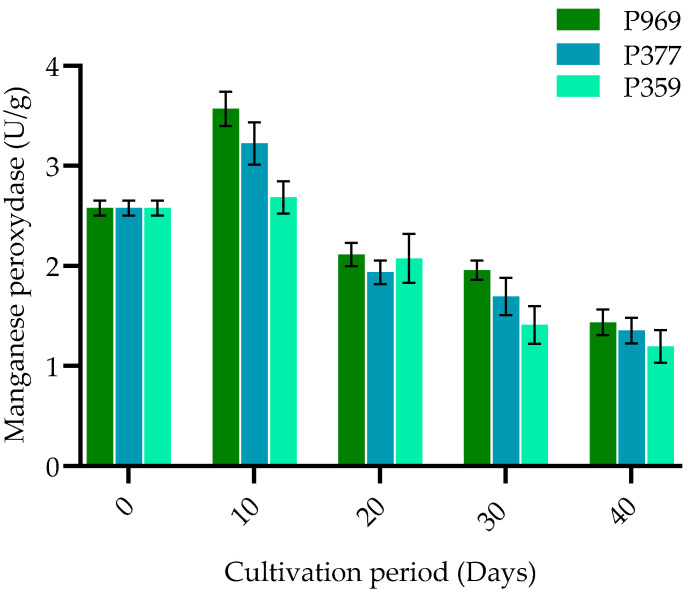
The bar plots show the change in manganese peroxidase activity in P969, P377, and P359 cultivated on pasteurized *C. fungigraminus* substrate. The black bars in the plots indicate the standard error of the mean. U/g: units per gram of the sample; P969: *Pleurotus ostreatus*; P377: *Pleurotus florida*; P359: *Pleurotus pulmonarius*.

**Figure 5 cimb-47-00143-f005:**
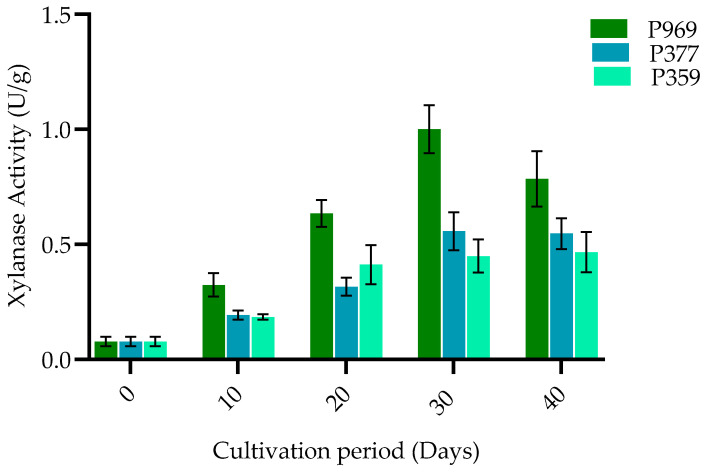
The bar plots show the change in xylanase activity in P969, P377, and P359 cultivated on pasteurized *C. fungigraminus* substrate. The black bars in the plots indicate the standard error of the mean. U/g: units per gram of the sample; P969: *Pleurotus ostreatus*; P377: *Pleurotus florida*; P359: *Pleurotus pulmonarius*.

**Table 1 cimb-47-00143-t001:** Mushroom species collected and used in this research.

No	Names	Strain Code	Geographic Origin
1	*Pleurotus florida*	P377	Yunnan
2	*Pleurotus ostreatus*	P969	Fujian
3	*Pleurotus pulmonarius*	P359	Fujian

**Table 2 cimb-47-00143-t002:** The mycelial development and yield potential of three species of *Pleurotus* grown on pasteurized *C. fungigraminus* substrates.

Mushroom Species	Mycelia Growth (Days)	Stipe Length (mm)	Pileus Diameter (mm)	Fresh Fruiting Body Weight (g)	BE (%)
*P*. *ostreatus*	25.33 ± 2.82 ^b^	35.94 ± 5.38 ^a^	70.05 ± 8.26 ^a^	164.50 ± 12.27 ^a^	78.23 ± 2.08 ^a^
*P*. *florida*	35.5 ± 2.50 ^a^	19.55 ± 9.10 ^b^	44.44 ± 11.85 ^b^	87.01 ± 10.50 ^c^	39.66 ± 5.50 ^c^
*P*. *pulmonarius*	29.33 ± 1.25 ^b^	36.20 ± 3.91 ^a^	63.73 ± 2.99 ^a^	117.60 ± 8.56 ^b^	59.88 ± 4.24 ^b^

Results expressed as mean ± standard deviation indicate significant differences at the 0.05 level (*p* ≤ 0.05) according to Duncan’s multiple range tests. Different letters (^a–c^) in the same column indicate significant differences at the 0.05 level (*p* ≤ 0.05).

**Table 3 cimb-47-00143-t003:** The nutrient composition in the fruiting bodies (mg·g^−1^ ± SD) of *Pleurotus* species grown on pasteurized *C. fungigraminus*.

Mushroom Species	Ash Content	Polysaccharide	Protein Content	Fiber Content	Fat Content
*P*. *ostreatus*	1.29 ± 0.10 ^b^	89.33 ± 2.01 ^b^	4.53 ± 0.01 ^a^	35.81 ± 0.34 ^a^	1.71 ± 0.09 ^b^
*P*. *pulmonarius*	1.15 ± 0.98 ^b^	92.29 ± 1.46 ^a^	2.99 ± 0.98 ^b^	30.42 ± 0.41 ^b^	2.01 ± 0.12 ^a^
*P*. *florida*	2.07 ± 0.07 ^a^	80.63 ± 2.06 ^c^	3.64 ± 0.17 ^b^	33.55 ± 0.02 ^b^	1.44 ± 0.27 ^b^

According to Duncan’s multiple range tests, results expressed as mean ± standard deviation indicate significant differences at the 0.05 level (*p* ≤ 0.05). Different letters (^a–c^) in the same column indicate significant differences at the 0.05 level (*p* ≤ 0.05).

**Table 4 cimb-47-00143-t004:** The concentration of heavy metals (mg·kg^−1^ ± SD) in the fruiting bodies of *Pleurotus* species grown in pasteurized *C. fungigraminus*.

Mushroom Species	Cadmium	Lead	Arsenic	Mercury
*P*. *ostreatus*	0.095 ± 0.012 ^a^	0.628 ± 0.031 ^a^	0.70 ± 0.056 ^a^	0.054 ± 0.002 ^a^
*P*. *pulmonarius*	0.136 ± 0.090 ^a^	0.018 ± 0.022 ^b^	0.047 ± 0.022 ^b^	0.012 ± 0.005 ^a^
*P*. *florida*	0.117 ± 0.001 ^a^	0.562 ± 0.015 ^a^	0.091 ± 0.054 ^a^	0.086 ± 0.001 ^a^

Outcomes presented as mean ± standard deviation and letters (^a,b^) in the same column show significant differences at the 0.05 level (*p* ≤ 0.05) based on Duncan’s multiple range tests.

## Data Availability

The datasets presented in this study will be made available upon request.

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
