# Peer review of "Enzymatic Activity and Nutrient Profile Assessment of Three *Pleurotus* Species Under Pasteurized *Cenchrus fungigraminus* Cultivation"

_cimb, 2025, doi:10.3390/cimb47030143_

Round 1
Reviewer 1 Report (Previous Reviewer 3)
Comments and Suggestions for Authors
The resubmitted manuscript described using pasteurized Cenchrus fungigraminus to cultivate the three Pleurotus sp. Their nutritional contents and five enzyme activities were investigated. Though doubts still existed because of the material and data issues, it would have been more reliable if the information provided by the authors had been believable. It is suggested that essential modifications be made to meet the quality of the Current Issues in Molecular Biology because the quality of the authors’ previous article (Ref 37, Niyimbabazi et al. 2022) was better than the current one. The major concerns are listed below.
1. The manuscript's title described the use of pasteurized Cenchrus fungigraminus to prepare the composted substrates. However, the pasteurization is not mentioned in Section 2.2.
2. The abstract described how the expression of other enzymes, in addition to laccase, varied between species and development. However, the description conflicted with the results shown in Figures 2 (amylase) and Figure 4 (manganese peroxidase). This may be because the authors modified their data in the last revised version, which was not the same as the first submission, and the unchanged descriptions remained in the abstract.
3. The enzyme activities for five enzymes were still not statistically analyzed, and the data had been drastically modified compared to the previous version. The significance of the data was also not given.
4. Table 4 didn’t show significant differences.
5. The results were still not compared with those using other materials. For example, C. fungigraminus was used to evaluate the enzymatic reactions of P. ostreatus in the authors’ previous publication (Ref 18, Claude et al. 2024), but the differences were not mentioned. The authors only described that the substrates had a notable impact on enzyme activity, although the activity varied at different developmental stages. It didn’t reveal meaningful interpretations.
6. Many references didn’t cite correctly. Some journal titles only showed single letters, for example, Refs 3 and 9. Some journal titles were wrong, for example, Refs 20 and 38. What was Ref 24?
7. Many typos could be found in the text.
Author Response
Thank you very much for taking the time to review this manuscript and providing useful comments. Kindly find the detailed responses below and resubmitted revised version.
Comments 1: The resubmitted manuscript described using pasteurized Cenchrus fungigraminus to cultivate the three Pleurotus sp. Their nutritional contents and five enzyme activities were investigated. Though doubts still existed because of the material and data issues, it would have been more reliable if the information provided by the authors had been believable. It is suggested that essential modifications be made to meet the quality of the Current Issues in Molecular Biology because the quality of the authors’ previous article (Ref 37, Niyimbabazi et al. 2022) was better than the current one. The major concerns are listed below: The manuscript's title described the use of pasteurized Cenchrus fungigraminus to prepare the composted substrates. However, the pasteurization is not mentioned in Section 2.2.
Response 1: Thank you for pointing these comments out. In this research, we utilized pasteurized Cenchrus fungigraminus as a growth substrate for various Pleurotus species. It is important to note that we did not employ pasteurized Cenchrus fungigraminus to prepare the composted substrate.
The pasteurized Cenchrus fungigraminus was done as follows: The grasses (Cenchrus fungigraminus) were soaked in regular water with the addition of 2% lime for 2 hours. After two hours of soaking, the grass was removed from the limed water and thoroughly pressed to remove excess water. Following this, 20% wheat bran and an additional 2% lime were incorporated based on the quantity of C. fungigraminus .
Based on this comment we have added (pasteurized Cenchrus fungigraminus). The change can be found in the Lines 108-109
Comments 2: The abstract described how the expression of other enzymes, in addition to laccase, varied between species and development. However, the description conflicted with the results shown in Figures 2 (amylase) and Figure 4 (manganese peroxidase). This may be because the authors modified their data in the last revised version, which was not the same as the first submission, and the unchanged descriptions remained in the abstract.
Response 2: Thank you for pointing out them. We agree with the comments. The abstract was revised. The changes can be found in the first page line 30-35.
Comments 3: The enzyme activities for five enzymes were still not statistically analyzed, and the data had been drastically modified compared to the previous version. The significance of the data was also not given.
Response 3: Thank you for pointing out them. Based on the previous comments from other reviewers’ we have considered the comments suggested, the errors were made before were corrected. That is why the resubmitted version was revised and we do appreciate comments from you and other reviewers.
Comments 4: Table 4 didn’t show significant differences.
Response 4: Thank you for pointing out the comment. The table was revised accordingly. The changes can be found in the line 254
Comments 5: The results were still not compared with those using other materials. For example, C. fungigraminus was used to evaluate the enzymatic reactions of P. ostreatus in the authors’ previous publication (Ref 18, Claude et al. 2024), but the differences were not mentioned. The authors only described that the substrates had a notable impact on enzyme activity, although the activity varied at different developmental stages. It didn’t reveal meaningful interpretations.
Response 5: Thank you for pointing out this. The results were compared with previous work by Claude et al. (2024). The changes can be found in the discussion section in the line 363-369.
Comments 6: Many references didn’t cite correctly. Some journal titles only showed single letters, for example, Refs 3 and 9. Some journal titles were wrong, for example, Refs 20 and 38. What was Ref 24?
Response 6: Thank you for pointing out this.
Ref 3 AJAHR. Is an abbreviation of Asian Journal of Agricultural and Horticultural Research as is mentioned on their offical website
Ref 9 revised as Agr.Rev.:Agriculture reviews is published under Agricultural Research Communication Centre journals. (ARCC Journals)
Ref 20 was corrected as Plant Var. Stud. Protect. Plant Varieties Studying and Protection journal.
Ref 38 and 24 were revised.
Comments 7: Many typos could be found in the text.
Response 7: Thank you for pointing out this. Typo errors were carefully revised in the manuscript.
Thank you very much

Reviewer 2 Report (Previous Reviewer 1)
Comments and Suggestions for Authors
Thanks to the authors for attending to the comments made. However, for the publication of the document, it is recommended to clarify what type of elisa was performed (direct, indirect), antibodies and antigen used for enzymatic quantification. As well as the commercial data in case it is a conventional kit.
Author Response
Thank you very much for taking the time to review this manuscript and providing useful comments. Kindly find the detailed responses below and resubmit the revised version.
Comments 1: Thanks to the authors for attending to the comments made. However, for the publication of the document, it is recommended to clarify what type of elisa was performed (direct, indirect), antibodies and antigens used for enzymatic quantification. As well as the commercial data in case it is a conventional kit.
Response 1: Thank you for pointing out these comments. The type of elisa performed in our study is indirect Elisa, as we utilized a commercially available kit designed for enzyme activity assays. The Elisa kits employed in our study were sourced from Shanghai Uplc-MS Testing Technology Co.Ltd.
Thank you very much.

Round 2
Reviewer 1 Report (Previous Reviewer 3)
Comments and Suggestions for Authors
The revised resubmitted manuscript described using pasteurized Cenchrus fungigraminus to cultivate the three Pleurotus sp. to investigate their nutritional contents and five enzyme activities. The questions remained, and the manuscript didn’t meet the quality of Current Issues in Molecular Biology. The major concerns are listed below.
1. The preparation of three Pleurotus sp. was still questioned. It was described that the second spawn were prepared in polyethylene bags filled with 500 g of sterile substrate, comprising 40% Miscanthus floridulus, 38% Dicranopteris dicholoma, 20% wheat bran, and 2% gypsum. What were the compositions by using pasteurized Cenchrus fungigraminus as the substrate?
2. As described above, what were the growth differences between the two substrates?
3. Section 2.3. It was described that the conventional Elisa (ELISA?) kits were sourced from Shanghai Uplc (UPLC?)-MS Testing Technology Co., Ltd. What were the kits’ product titles and numbers?
4. The abstract described that “P. ostreatus was faster than others.” What did it mean?
5. The abstract described “P. ostreatus inconsistent with manganese peroxidase, which showed the highest pick on the 10th day among P. florida, P. pulmonarius”. What did it mean?
6. The enzyme activities for five enzymes were still not statistically analyzed, and the data had been drastically modified compared to the previous version. The significance of the data was also not given.
7. The results were still not compared with those using other materials. For example, C. fungigraminus was used to evaluate the enzymatic reactions of P. ostreatus in the authors’ previous publication (Ref 18, Claude et al. 2024), but the differences were not mentioned. The authors only described that the substrates had a notable impact on enzyme activity, although the activity varied at different developmental stages. It didn’t reveal meaningful interpretations. How did the readers know the results indicated the usefulness of pasteurized C. fungigraminus for cultivating Pleurotus species and expression enzyme activity in different Pleurotus species, as described in the abstract?
8. Section 3.2. What are the standards for determining heavy metals concentration to meet food safety and regulations?
9. The number of decimal places in the tables was different. For example, the values shown for P. florida in Table 2, the value shown for P. pulmonarius in Table 3, and the value shown for P. ostreatus in Table 4.
10. Section 3.3. It was described that on the thirtieth day, the xylanase enzyme activity of P969 reached its peak at the fruiting body stage, measuring (1.3 U/g). However, the value was different when compared with that in Figure 5.
11. Many references still needed to be cited correctly. Some journal titles only showed single letters, for example, Refs 3, 4, 15…… Some journal titles used lowercase letters, for example, Ref 10. Some used full titles, for example, Ref 7.
12. Many more are not listed……
Author Response
Thank you very much for taking the time to review this manuscript and providing useful comments. Kindly find the detailed responses below and resubmit the revised version.
Comments 1: The preparation of three Pleurotus sp. was still questioned. It was described that the second spawn were prepared in polyethylene bags filled with 500 g of sterile substrate, comprising 40% Miscanthus floridulus, 38% Dicranopteris dicholoma, 20% wheat bran, and 2% gypsum. What were the compositions by using pasteurized Cenchrus fungigraminus as the substrate?
Response 1: Thank you for pointing these comments out. In the section 2.1 shows the preparation of primary and secondary spawns. This formulation of 40% Miscanthus floridulus, 38% Dicranopteris dicholoma, 20% wheat bran, and 2% gypsum was used to prepare the second spawn which is the seed for the next step and described in section 2.2. The prepared second spawn was used to inoculate the prepared substrate (pasteurized Cenchrus fungigraminus) for growing three pleurotus species.
Comments 2: As described above, what were the growth differences between the two substrates?
Response 2: Thank you for pointing out them. The method for preparing secondary spawns differs from substrate preparation for pleurotus mushroom cultivation. In fact, section 2.1 shows the source of the seeds or spawns we used in this research.
Comments 3: Section 2.3. It was described that the conventional Elisa (ELISA?) kits were sourced from Shanghai Uplc (UPLC?)-MS Testing Technology Co., Ltd. What were the kits’ product titles and numbers?
Response 3: Thank you for pointing out them. The boxes were titled Sinobestbio enzyme assay kits. Each enzyme has its box essay kit, and unfortunately, the lot number was not recorded. After finishing the experiment, I cleaned all the trash and threw them away.
Those are some unprofessional pictures taken while i was doing experiments
Comments 4: The abstract described that “P. ostreatus was faster than others.” What did it mean?
Response 4: Thank you for pointing out them. This means the mycelium growth rate of P. ostreatus was faster than other pleurotus species (P.florida and P. pulmonarius).The changes can be found in the line 26-28
Comments 5: The abstract described “P. ostreatus inconsistent with manganese peroxidase, which showed the highest pick on the 10th day among P. florida, P. pulmonarius”. What did it mean?
Response 5: Thank you for pointing out them. Laccase exhibited its peak activity of 13.8 U/g on the twentieth during the growth phase during the growth phase and gradually decreased to the fruiting body stage in P. ostreatus. In contrast, manganese peroxidase reached its highest activity on the tenth of the growth phase, also decreasing gradually to the fruiting body, particularly when compared to P. florida and P. pulmonarius. The changes can be found in line 33.
Comments 6: The enzyme activities for five enzymes were still not statistically analyzed, and the data had been drastically modified compared to the previous version. The significance of the data was also not given.
Response 6: Thank you for pointing out them. As you may have noticed, the previous version has been replaced with the resubmitted one. It is important to evaluate the revised version on its own merits, rather than comparing it to the old version.
Comments 7: The results were still not compared with those using other materials. For example, C. fungigraminus was used to evaluate the enzymatic reactions of P. ostreatus in the authors’ previous publication (Ref 18, Claude et al. 2024), but the differences were not mentioned. The authors only described that the substrates had a notable impact on enzyme activity, although the activity varied at different developmental stages. It didn’t reveal meaningful interpretations.
Response 7: Thank you for pointing out them. In the previous article, you said in the comment, I served as the second author alongside the first author., in that research we have used different substrates (Saccharum arundinaceum , Cenchrus fungigraminus, and Miscanthus floridulus) and different formula to grow pleurotus. The different enzymes were determined by comparing their activity across various substrates. However, this research focuses on the changes in enzyme activity among different Pleurotus species at various developmental stages. The considerable differences in enzyme activity among the Pleurotus species observed in this research were performed in all different developmental stages. However, Pleurotus ostreatus shows higher efficient activity compared to other species grown in C. fungigraminus. Research indicates that the conversion of lignocelluloses into soluble sugars by Pleurotus species is significantly influenced by various nonspecific oxidative enzymatic systems, primarily laccases and manganese peroxidases those findings were in the align with previous work by Claude et al. (2024), which demonstrated that the growth of Pleurotus ostreatus on different substrates such as C. fungigraminus significantly impacted enzyme activity, although this activity varied at different stages of development. The changes can be found in the discussion section in the line 370-378.
Comments 8: Section 3.2. What are the standards for determining heavy metals concentration to meet food safety and regulations?
Response 8: Thank you for pointing out the comment. Determining heavy metal concentration to meet food safety and regulations was done based on the WHO/FAO Expert Committee on Food Additives (JECFA) as follows: Cd: 1 mg/kg, Ld:2mg/kg, As:3mg/kg, Hg:1mg/kg.
Comments 9: The number of decimal places in the tables was different. For example, the values shown for P. florida in Table 2, the value shown for P. pulmonarius in Table 3, and the value shown for P. ostreatus in Table 4.
Response 9: Thank you for pointing out the comment. Considering the useful comments and advise all tables 2,3,4, were revised accordingly.
Comments 10: Section 3.3. It was described that on the thirtieth day, the xylanase enzyme activity of P969 reached its peak at the fruiting body stage, measuring (1.3 U/g). However, the value was different when compared with that in Figure 5.
Response 10: Thank you for pointing out the comment. Measuring were numbers were revised based on the useful comments. The changes can be found in the discussion section in the line 330.
Comments 11: Many references still needed to be cited correctly. Some journal titles only showed single letters, for example, Refs 3, 4, 15…… Some journal titles used lowercase letters, for example, Ref 10. Some used full titles, for example, Ref 7.
Response 11: References 3, 4, 7, 10, and 15, as well as the entire references section, have been thoroughly revised.
Thank you very much

Round 3
Reviewer 1 Report (Previous Reviewer 3)
Comments and Suggestions for Authors
The 2nd revised resubmitted manuscript described using pasteurized Cenchrus fungigraminus to cultivate the three Pleurotus sp. to investigate their nutritional contents and five enzyme activities. Many questions remained, and the manuscript didn’t meet the quality of Current Issues in Molecular Biology. The major concerns are listed below.
1. The preparation of three Pleurotus sp. was still questioned. It was described that the second spawn were prepared in polyethylene bags filled with 500 g of sterile substrate, comprising 40% Miscanthus floridulus, 38% Dicranopteris dicholoma, 20% wheat bran, and 2% gypsum. What were the compositions by using pasteurized Cenchrus fungigraminus as the substrate? The authors still didn’t describe it in Section 2.2.
2. As described above, what were the growth differences between the two substrates? The authors still didn’t describe it in the manuscript.
3. Section 2.3. It was described that the conventional Elisa (ELISA?) kits were sourced from Shanghai Uplc (UPLC?)-MS Testing Technology Co., Ltd. What were the kits’ product titles and numbers? The authors’ reply was questioned.
4. The abstract described “inconsistent with manganese peroxidase which showed the highest pickpeak on the 10th day among P. florida, P. pulmonarius while the expression of other enzymes varied between species and developmental phases”. What did it mean?
5. The enzyme activities for five enzymes were still not statistically analyzed, though the data had been drastically modified compared to the previous version. The significance of the data was also not given, and the authors still didn’t describe it in the manuscript.
6. The results were still not compared with those using other materials. For example, C. fungigraminus was used to evaluate the enzymatic reactions of P. ostreatus in the authors’ previous publication (Ref 18, Claude et al. 2024), but the differences were not mentioned. The authors only described that the substrates had a notable impact on enzyme activity, although the activity varied at different developmental stages. It didn’t reveal meaningful interpretations. How did the readers know the results indicated the usefulness of pasteurized C. fungigraminus for cultivating Pleurotus species and expression enzyme activity in different Pleurotus species, as described in the abstract?
7. Section 3.2. What are the standards for determining heavy metals concentration to meet food safety and regulations? The authors replied but didn’t describe them in the manuscript, and the references should be cited.
8. The authors modified the number of decimal places in the tables to be the same, but they should be shown according to the significant digits related to the methods and instruments used. They may not be the same. In addition, what was 25.33.9±2.82 for the mycelium growth of P. ostreatus in Table 2? Were they correct to show 0.05±0.00, 0.01±0.00, and 0.08±0.00 for the mercury concentration in Table 4?
9. Section 3.3. It was described that on the thirtieth day, the xylanase enzyme activity of P969 reached its peak at the fruiting body stage, measuring (1.3 U/g). However, the value was different when compared with that in Figure 5. The authors modified it, and the value was 1.1 U/g. Was it convincing data?
10. Many references still need to be cited correctly. Some journal titles only showed single letters, for example, Refs 18, 21, 26…… Some didn’t provide pages or article numbers, for example, Refs 4 and 9.
11. Many more are not listed……
Author Response
Cover letter to reviewer’s comments
Thank you very much for taking the time to review this manuscript and providing useful comments. Kindly find the detailed responses below and resubmit the revised version.
Comments 1: The preparation of three Pleurotus sp. was still questioned. It was described that the second spawn were prepared in polyethylene bags filled with 500 g of sterile substrate, comprising 40% Miscanthus floridulus, 38% Dicranopteris dicholoma, 20% wheat bran, and 2% gypsum. What were the compositions by using pasteurised Cenchrus fungigraminus as the substrate? The authors still didn’t describe it in Section 2.2.
Response 1: Thank you for pointing this comment out. The pasteurised Cenchrus fungigraminus grasses were supplemented, with 20% wheat bran, and 2% lime. The polyethylene bags containing 500 grams of the substrate were directly inoculated with secondary spawn, as detailed in the main text in section 2.2. line 113-117
Comments 2: As described above, what were the growth differences between the two substrates? The authors still didn’t describe it in the manuscript.
Response 2: Thank you for pointing this comment out. The table below describes the differences between two substrates
|
No |
Substrate |
Use for |
Disinfection method |
Role |
Cultivation |
|
1 |
40% Miscanthus floridulus, 38% Dicranopteris dicholoma, 20% wheat bran, and 2% gypsum. |
Second spawn preparation |
Sterilized in the autoclave machine at 121oC for 120 minutes. Cooled down for 24hrs then inoculated by primary spawn |
After mycelium colonization used to inoculate the main substrate pasteurized Cenchrus fungigraminus |
Not cultivated, Used as spawn or seed. |
|
2 |
Pasteurized Cenchrus fungigraminus |
The main substrate used in this research |
Pasterilized Cenchrus fungigraminus grasses were soaked in the water with the addition of 2% lime for 2 hours. After two hours of soaking, the grass was removed from the limed water and thoroughly pressed to remove excess water. After that, the grasses were supplemented, with 20% wheat bran,2% lime. Then inoculated by secondary spawn.
|
After the total mycelium conization bags are transferred to the cultivation room for mushroom cultivation. |
Cultivated for yielding (Fruiting body) |
Comments 3: Section 2.3. It was described that the conventional Elisa (ELISA?) kits were sourced from Shanghai Uplc (UPLC?)-MS Testing Technology Co., Ltd. What were the kits’ product titles and numbers? The authors’ reply was questioned.
Response 3: Thank you for pointing this comment out. The boxes were titled Sinobestbio enzyme assay kits. Each enzyme has its box essay kit, and unfortunately, the lot number was not recorded. After finishing the experiment, the author cleaned all the trash and threw them away.
Here are some unprofessional pictures taken while conducting experiments.
The authors confirm that the information provided in this comment is accurate to the best of our knowledge.
Comments 4: The abstract described “inconsistent with manganese peroxidase which showed the highest pickpeak on the 10th day among P. florida, and P. pulmonarius while the expression of other enzymes varied between species and developmental phases”. What did it mean?
Response 4: Thank you for pointing this comment out. The expression of manganese peroxidase in P.ostreatus reached its highest activity 3.6 U/g on the 10th day compared to P. florida, and P. pulmonarius. The expression of other enzymes varied between species and developmental phases. The changes can be found in lines 32-34
Comments 5: The enzyme activities for five enzymes were still not statistically analyzed, though the data had been drastically modified compared to the previous version. The significance of the data was also not given, and the authors still didn’t describe it in the manuscript.
Response 5: Thank you for pointing this comment out. As you may have noticed, the previous version has been replaced with the revised manuscript based on the comments of reviewers and was resubmitted. It is important to evaluate the revised version on its own merits, rather than comparing it to the old version. Each enzyme activity in different pleurotus species was analysed and were described in the result section.
Comments 6: The results were still not compared with those using other materials. For example, C. fungigraminus was used to evaluate the enzymatic reactions of P. ostreatus in the authors’ previous publication (Ref 18, Claude et al. 2024), but the differences were not mentioned. The authors only described that the substrates had a notable impact on enzyme activity, although the activity varied at different developmental stages. It didn’t reveal meaningful interpretations. How did the readers know the results indicated the usefulness of pasteurized C. fungigraminus for cultivating Pleurotus species and expression enzyme activity in different Pleurotus species, as described in the abstract?
Response 6: Thank you for pointing out them. In Claude et al. 2024 article mentioned in the reviewer’s comment, I served as the second author alongside the first author., in this article we used different substrates, the formula used, and disinfection methods were different to grow pleurotus ostreatus. Enzyme activities were compared in groups. The details were described in the discussion section. The changes can be found in the discussion section in the line 370-378.
Comments 7: Section 3.2. What are the standards for determining heavy metals concentration to meet food safety and regulations? The authors replied but didn’t describe them in the manuscript, and the references should be cited.
Response 7: Thank you for pointing this comment out. The revised and cited in the manuscript. The changes can be found in the discussion section in the line 253-254, and 500-502.
Comments 8: The authors modified the number of decimal places in the tables to be the same, but they should be shown according to the significant digits related to the methods and instruments used. They may not be the same. In addition, what was 25.33.9±2.82 for the mycelium growth of P. ostreatus in Table 2? Were they correct to show 0.05±0.00, 0.01±0.00, and 0.08±0.00 for the mercury concentration in Table 4?
Response 8: Thank you for pointing this comment out. The tables were revised. Table 4 presents the concentration of heavy metals in various Pleurotus species. Based on the corresponding author's advice, Table 4 should have three decimal places due to the number of data are small in all repeats. Ranged of 0. mg·kg-1 up to 1mg·kg-1. Data from three oyster mushrooms were analyzed using a one-way ANOVA test in SPSS software version 27 to compare the means of multiple groups. A post hoc analysis was conducted using the Duncan test at the 5% significance level to identify significant differences between group means.
Comments 9: Section 3.3. It was described that on the thirtieth day, the xylanase enzyme activity of P969 reached its peak at the fruiting body stage, measuring (1.3 U/g). However, the value was different when compared with that in Figure 5. The authors modified it, and the value was 1.1 U/g. Was it convincing data?
Response 9: Thank you for pointing this comment out. Yes, it is
Comments 10. Many references still need to be cited correctly. Some journal titles only showed single letters, for example, Refs 18, 21, 26…… Some didn’t provide pages or article numbers, for example, Refs 4 and 9.
Response 10: References 18,21,26,4,9 and the entire references section, have been thoroughly revised.

This manuscript is a resubmission of an earlier submission. The following is a list of the peer review reports and author responses from that submission.
Round 1
Reviewer 1 Report
Comments and Suggestions for Authors
Las estrategias para mejorar los sistemas de producción son de gran interés para desarrollar estrategias de optimización de procesos. Por ello, esta investigación es de interés científico. Sin embargo, para su publicación se requiere la atención de las siguientes observaciones:
1. Revisión de la gramática y el lenguaje del manuscrito.
2. Se recomienda que la Tabla 1 permanezca en la misma página.
3. En el apartado de metodología no queda claro el volumen de sustrato colocado en cada una de las bolsas.
4. Se recomienda separar la unidad del valor numérico.
5. Se recomienda verificar que los nombres científicos estén en cursa.
6. Página 3, Línea 81 92. Revisar y corregir.
Página 4, línea 103: Se recomienda revisar la redacción.
8. En el apartado de materiales y métodos se recomienda incluir alguna metodología que indique la calidad microbiana del sustrato, es decir, que permita incluir en los resultados la eficiencia del proceso.
9. En las tablas 2, 3, 4 se recomienda revisar los literales de los promedios, que sean correctos y congruentes con los valores numéricos.
10. La tabla 2 requiere una nota al pie de tabla.
11. Página 5, líneas 164-167. Revise el formato de superíndice de la expresión mg kg -1
12. Revise los datos de desviación estándar en las figuras y observe que en algunos casos es mayor que el valor medio.
13. En la discusión, revise la redacción de los dos primeros párrafos. También es necesario mejorarla.
14. La metodología para la cuantificación de enzimas no es clara y no se entiende por qué no se presentan los datos de lacasa.
15. Se recomienda revisar el índice de similitud (43%) y cuidar la autocitación.

Se requiere revisión del idioma.
Author Response
|
Response to Reviewer Comments
|
||
|
1. Summary |
|
|
|
Thank you very much for taking the time to review this manuscript. Kindly find the detailed responses below and the submitted revised version. Comments 1: Review of the manuscript's grammar and language. Response 1: All Comments highlighted in the PDF files related to grammar and language were revised accordingly. The revised version is uploaded in the system. Thank you for pointing out them. We agree with the comments. Comments 2: It is recommended that Table 1 remain on the same page. Response 2: Thank you for pointing out them. We agree with the comments. The table was fixed in the same page. The changes can be found on page 3, paragraph 1 line 103. Comments 3: In the methodology section, the volume of substrate placed in each bag is not clear. Response 3: Thank you for pointing out them. We agree with the comments. The volume of the substrate in each bag is 500g/per bag. The changes can be found on page 3, paragraph 2 line 112. Comment 4: It is recommended to separate the unit from the numerical value. Response 4: Thank you for pointing them out. We agree with the comments. The unit was separated into whole revised versions. Different corrections can be found on pages 3, 4, paragraphs 2, 4, and 1, in lines 118, 110, 135,136, 137, 138, 149… Comment 5: It is recommended to verify that scientific names are in italics. Response 5: Thank you for pointing them out. We agree with the comments. The scientific names were revised and put in italics in the whole manuscript. The change can be found on pages 3 and 4, paragraphs 1, and 3, lines 97, 131, 170 and in the whole manuscript. Comment 6: Page 3, Lines 81-92: Review and correct. Page 4, Line 103: It is recommended to review the wording. Response 6: Thank you for pointing them out. We. agree with the comments. Page 3,4 were revised. The change can be found on pages 3, and 4, paragraphs 1, and 3, lines 97, 131, 170 Comment 7: In the materials and methods section, it is recommended to include a methodology that indicates the microbial quality of the substrate, that is, to allow inclusion of the process efficiency in the results. Response 7: Thank you for pointing them out. We agree with the comments. The materials and methods section was revised according to the recommendation of reviewers. Comment 8: In Tables 2, 3, and 4, it is recommended to review the literals of the averages, ensuring they are correct and congruent with the numerical values. Response 8: Thank you for pointing them out. We agree with the comments. Tables 2, 3, and 4 were revised. The change can be found on pages 5,6, paragraph 2,1,3, lines212,233,251. Comment 9: Table 2 requires a footnote. Response 9: Thank you for pointing them out. We agree with the comments. Tables 2 were revised. The change can be found on page 6, paragraph 1, lines 233. Comment 10: Review the superscript format of the expression mg kg -1 Response 10: Thank you for pointing them out. We agree with the comments. The superscript format of the expression mg kg -1 was carefully revised. The change can be found on page 6, paragraph 1,2,3,4, lines 233,243,244,245,251. Comment 11: Review the standard deviation data in the figures and note that in some cases it is greater than the mean value. Response 11: Thank you for pointing them out. We agree with the comments. The standard deviation data in the figures were carefully revised. The change can be found on page 7,8,9, Figure 1,2,3,4,5 Comment 12: In the discussion, review the wording of the first two paragraphs. It is also necessary to improve it. Response 12: Thank you for pointing them out. We agree with the comments. The discussion section was well improved. The change can be found on page 10, lines 329-366 Comment 13: The methodology for enzyme quantification is not clear, and it is not understood why laccase data are not presented. Response 13: Thank you for pointing them out. We agree with the comments. The methodology section for the enzyme activity was well improved. The change can be found on page 3,4, lines 128-166
|
||

Reviewer 2 Report
Comments and Suggestions for Authors
The study assessed the growth rates, yields and enzyme activities of three Pleurotus species namely Pleurotus ostreatus, Pleurotus florida, and Pleurotus pulmonarius cultivated on non-sterilized giant grass substrate.
I found the paper to be well written, without the problems of scientific expression.
The analysis methods are detailed, and the experimental design is correct.
The results are processed statistically, presented in tables and graphically,
The discussions are detailed, clearly expressed, and scientific, following similar research.
The bibliographic resources are relevant for research.
Minor observation
I did not find the botanical name of giant junco grass anywhere in the manuscript. It should be specified.
Author Response
|
Response to Reviewer Comments
|
||
|
1. Summary |
|
|
|
Thank you very much for taking the time to review this manuscript. Kindly find the detailed responses below and the submitted revised version. Comments 1: I did not find the botanical name of giant junco grass anywhere in the manuscript. It should be specified. Review of the manuscript's grammar and language. Response 1: Thank you for pointing out this comment. We agree with it. The scientific of giant juncao grass is Cenchrus fungigraminus and was used in the revised manuscript. we can find it in the WFO plant list. Link: http://www.worldfloraonline.org/taxon/wfo-1000049395; the author is Prof Lin Zhanxi. The nomenclature reference can be found on the international plant name index. Link: https://www.ipni.org/n/urn:lsid:ipni.org:names:77320785-1 .
|
||

Reviewer 3 Report
Comments and Suggestions for Authors
The manuscript described using non-sterilized giant grass to cultivate three Pleurotus species, including P. ostreatus, P. florida and P. pulmonarius, and investigating their growth rate, yields, and enzyme activities. The manuscript is not recommended for Current Issues in Molecular Biology for the following reasons.
1. The introduction was too brief. It seems the use of giant grass as the material to cultivate was not a novel idea and has been studied, including the authors’ previous publication (Ref 5, Niyimbabazi et al. 2022). Indeed, these two works were very similar.
2. The source of the giant grass was unknown.
3. The statistical analysis was described in section 3. However, it should be in section 2.5. In addition, the enzyme activities for five enzymes were not statistically analyzed.
4. There were two Figure 4.
5. The importance of using giant grass was not revealed, and the results were also not compared with those using other materials.
6. The discussion was poorly described.
7. The citations in the text didn’t follow the journal’s rule.
Author Response
|
Response to Reviewer comments
|
||
|
1. Summary |
|
|
|
Thank you very much for taking the time to review this manuscript. Kindly find the detailed responses below and the submitted revised version. Comments 1: The introduction was too brief. It seems the use of giant grass as the material to cultivate was not a novel idea and has been studied, including the authors’ previous publication (Ref 5, Niyimbabazi et al. 2022). Indeed, these two works were very similar. Response 1: Thank you for pointing out them. We agree with the comments. The introduction has been revised and enhanced in the updated manuscript. Giant grass was used for fungi cultivation and is scientifically named Cenchrus fungigraminus. The previous author referenced in the comments utilized a composted substrate, which differs from the one employed in this article. Comments 2: The source of the giant grass was unknown. Response 2: Thank you for pointing out this comment. We agree with it. The scientific of giant juncao grass is Cenchrus fungigraminus and was used in the revised manuscript. We can find it in the WFO plant list. Link: http://www.worldfloraonline.org/taxon/wfo-1000049395; the author is Prof Lin Zhanxi. The nomenclature reference can be found in the international plant name index. Link: https://www.ipni.org/n/urn:lsid:ipni.org:names:77320785-1 Comments 3: The statistical analysis was described in section 3. However, it should be in section 2.5. In addition, the enzyme activities for five enzymes were not statistically analyzed. Response 3: Thank you for pointing them out. We agree with the comments. The statistical analysis section was revised accordingly. The five enzyme activities were analysed and revised carefully. The change can be found on page 5 line 190; Pages 7,8,9, Figure 1,2,3,4,5 Comments 4: There were two Figure 4. Response 4: Thank you for pointing it out. We agree with the comment. The figures were well-revised. The change can be found on page 9, Figure 4,5 Comments 5: The importance of using giant grass was not revealed, and the results were also not compared with those using other materials. Response 5: Thank you for pointing it out. We agree with the comments. The importance of using giant grass (Cenchrus fungigraminus )was detailed in the introduction part of this manuscript Cenchrus fungigraminus is a perennial grass with high cellulose and crude protein content, capable of yielding up to 200 tons per hectare as biomass, it has abilities to grow at a wide range of temperatures utilizing various lignocelluloses and the. The change can be found on page 2 paragraph 3 Lines 56-59. Comments 6: The discussion was poorly described. Response 6: Thank you for pointing them out. We agree with the comments. The discussion section was well improved. The change can be found on page 10, lines 329-366 Comments 7: The citations in the text didn’t follow the journal’s rule Response 7: Thank you for pointing it out. We agree with the comments. The citations were revised according to the journal’s rules. The change can be found in the revised manuscript.
|
||
|
|
||

Reviewer 4 Report
Comments and Suggestions for Authors
Comments and Suggestions for Authors were added as an attachment
Review Report
Journal: Current Issues in Molecular Biology (ISSN 1467-3045)
Title: Enzymatic activity and nutrient profile assessment of three Pleurotus species under non-sterilized giant grass cultivation.
In present study Giant non-sterilised grass was used as a substrate to grow three mushroom species, Pleurotus ostreatus, Pleurotus florida and Pleurotus pulmonaius. Authors chose the aim of their work - to evaluate their growth rate, yields, and enzyme activities. It seems to be interesting. Nevertheless some points in the manuscript must be better explained and some corrections and revisions are needed.
The introduction is too short. The title of the article begins with the words “Enzymatic activity and….” which indicates the main issue of the presented work. Unfortunately there is no reference to enzymes in the introduction. In the presented work, the author describes the activity of five enzymes about which there is no word in the introduction, lacking any purpose or explanation of the decision to choose these particular 5 enzymes. The last two sentences of the introduction indicate that the purpose of the work is to use non-sterilized substrate in the cultivation of mushrooms. This is not consistent with the title - it misleads the reader. This should be completed or the title should be changed. I positively evaluate the description of mushroom cultivation and the methodology for measuring activity.
In line 41: There should be a reference to Table 1
In line 72, 73: I suggest list the enzymes in the order they are described below
In line 110: I recommend adding a description of BCA reagent
In line 121: Repetition of the word ‘three’
In line 129, 130, 222,: I do not see the need to repeat the previously introduced information that Pleurotus ostroatus is P969, etc. It is unnecessary
In line 132-135, 150-152, 163, 165, 166, 210, 222, 249, 251, 260: Please decide how to write the name of the fungus, e.g. the full name 'Pleurotus strefatus' or the shortened form 'P. strefatus'. It is good practice to write the full name when it first appears in the text and then use only one form.
In line 164, 166, 167: No superscript in the unit
In line 188: Why does the name ‘Pleurotus’ start with a lower case letter?
In line 180-183 and 192-197: these are sentences that sound the same - such a practice is in poor taste
In line 184: All enzyme activities are expressed in Unit per gram. There is no information per gram of what?
In line 208: Why new paragraph? the content is about the same topic
In line 210: Why is the fragment ‘...the highest peak (1.6 U/g)...’ written in italics?
In line 218: Why does the name ‘Manganese’ start with a capital letter?
In line 220: The word 'enzyme' is unnecessary
In line 223: No spaces in the fragment: ‘…2,6U/g…’
In line 225: Missing full stop at the end of the sentence
In line 231: A lonely ‘P’ at the end of the line
In line 234: Why does the name ‘Xylanase’ start with a capital letter?
In line 237: Not very well thought out sentence
In line 244, 245: Wrong way to refer to literature
In line 249: Value in brackets should be separated by a space
In line 251: No spaces in the fragment: ‘…2.05mg.g-1 …’
In line 252: ‘Miah et el’ should be ‘Miah et al.’
In line 259: Too many spaces in the fragment: ‘…reported by Dedousi et al. …’; No spaces in the fragment: ‘…Dedousi et al.(2024)…’
In line 317: ‘Gowda N N’ or ‘Gowda N’
· Please standardize the form of units. The author uses the ‘U/g’ notation once and ‘mg.g-1‘ the next time.
· Charts on Figures 1-4 have fonts that are too large compared to the font in the text. In the captions of the Figures 1-4 we can again find that Pleurotus ostroatus is P969, etc. It is unnecessary. Figure 1 caption: Why does the name ‘Laccase’ start with a capital letter? In the figure you can see an increase in enzyme activity. Why wasn't the experiment extended to assess its maximum?
· The most difficult point to understand is the fragment (line 222) where the author writes that the highest peak activity occurs on the 10th day of cultivation. Analyzing graph 4 we cannot agree with this. Explanation needed.
The technical quality and readability of the description are not enough for “Current Issues in Molecular Biology” in its current version.

Author Response
|
1. Summary |
|
|
|
Thank you very much for taking the time to review this manuscript. Kindly find the detailed responses below and the submitted revised version. Comments 1: The introduction is too short. The title of the article begins with the words “Enzymatic activity and….” which indicates the main issue of the presented work. Unfortunately there is no reference to enzymes in the introduction. In the presented work, the author describes the activity of five enzymes about which there is no word in the introduction, lacking any purpose or explanation of the decision to choose these particular 5 enzymes. The last two sentences of the introduction indicate that the purpose of the work is to use non-sterilized substrate in the cultivation of mushrooms. This is not consistent with the title - it misleads the reader. This should be completed, or the title should be changed. I positively evaluate the description of mushroom cultivation and the methodology for measuring activity. Response 1: Thank you for pointing out them. We agree with the comments. The introduction has been revised and enhanced in the updated manuscript. The enzyme activity was referred to in the introduction part in the introduction. Mushroom produces many extracellular enzymes that can degrade lignocellulose materials into soluble compounds of a relatively low molecular mass those e soluble compounds have remarkable nutritional value, and the whiterot fungi are the most ligninolytic microorganisms in nature. Decomposition is performed by a complex set of enzymes, containing lignin peroxidase, Laccase, and Xylanase, involves microbes for the solubilization of insoluble macromolecules like cellulose, hemicelluloses, lignin, based on the secretions of the enzymes in their substrate.The last paragraph of the introduction part was completed in the line of title. indicated the purpose of the work. The change can be found on page 2 paragraph line 4, lines 68-92. Comments 2: In line 41: There should be a reference to Table 1 Response 2: Thank you for pointing it out. We agree with this comment. A reference related to Table 1 was added. The change can be found on page 3, paragraph 1, line 101. Comments 3: In lines 72, 73: I suggest list the enzymes in the order they are described below Response 3: Thank you for pointing it out. We agree with this comment. The list of the five enzymes has been organized in alignment with the five analyzed figures. The changes can be referenced on page 3, paragraph 3, lines 131, and 132, as well as on pages 7, 8, and 9 in figures 1, 2, 3, 4, and 5. Comments 3: In line 110: I recommend adding a description of the BCA reagent Response 3: Thank you for pointing it out. We agree with this comment. The description of the BCA reagent was added (Bicinchoninic Acid protein assay kit). Reagent A is a working reagent composed of bicinchoninic acid sodium salt. Reagent B is a copper Solution composed of sulfate pentahydrate. The BCA assay is commonly used to determine protein concentration in mushroom samples. The changes can be referenced on page 4, paragraph 4, line 181. Comments 4: In line 121: Repetition of the word ‘three’ Response 4: Thank you for pointing it out. We agree with this comment. The repetition of three was corrected. The changes can be referenced on page 5, paragraph 1, line 193. Comments 5: In line 129, 130, 222,: I do not see the need to repeat the previously introduced information that Pleurotus ostroatus is P969, etc. It is unnecessary Response 5: Thank you for pointing it out. We agree with this comment. The repeat of Pleurotus strains was revised in the whole revised manuscript. The changes can be found on page 5, paragraph 2, lines 201-208. Comments 6: In line 132-135, 150-152, 163, 165, 166, 210, 222, 249, 251, 260: Please decide how to write the name of the fungus, e.g. the full name 'Pleurotus strefatus' or the shortened form 'P. strefatus'. It is good practice to write the full name when it first appears in the text and then use only one form. Response 6: Thank you for pointing it out. We agree with this comment. The name of the fungus was revised and decided to use the shortened form as advised by the reviewer. The changes can be found on page 5,6, lines 202-205; 2023-225;224-226. Comments 7: In lines 164, 166, 167: No superscript in the unit Response 7: Thank you for pointing it out. We agree with this comment. The units were revised. The changes can be found on page 6, lines 235,245-247, 253. Comments 8: In line 188: Why does the name ‘Pleurotus’ start with a lowercase letter? Response 8: Thank you for pointing it out. We agree with this comment. The scientific name of the fungus was revised by starting with a capital letter. The changes can be found on page 7, line 275. Comments 9: In line 180-183 and 192-197: these are sentences that sound the same - such a practice is in poor taste Response 9: Thank you for pointing it out. We agree with this comment. The sentences were revised in the manuscript. The changes can be found on page 6, lines 265-268. Comments 10: All enzyme activities are expressed in Units per gram. There is no information per gram of what? Response 10: Thank you for pointing it out. We agree with this comment. The text now includes a unit per gram of sample (μmol oxidised or released /min/g). The changes can be found on pages 7, 8, and 9, lines 274,2,88, 301, 316, and 331. Comments 11: In line 208: Why new paragraph? The content is about the same topic Response 11: Thank you for pointing it out. We agree with this comment. The paragraphs were well-revised. The changes can be found on page 8, Paragraph 1, line 290 Comments 12: In line 210: Why is the fragment ‘...the highest peak (1.6 U/g)...’ written in italics? Response 12: Thank you for pointing it out. We agree with this comment. The fragment was well-revised. The changes can be found on page 8, Paragraph 1, line 293. Comments 13: In line 218: Why does the name ‘Manganese’ start with a capital letter? Response 13: Thank you for pointing it out. We agree with this comment. The manganese was corrected. The changes can be found on page 8, Paragraph 2, line 303. Comments 14: In line 220: The word 'enzyme' is unnecessary Response 14: Thank you for pointing it out. We agree with this comment. The word enzyme was removed as is advised. The changes can be found on page 8, Paragraph 2, line 305. Comments 15: In line 223: No spaces in the fragment: ‘…2,6U/g…’ Response 15: Thank you for pointing it out. We agree with this comment. The segment was revised as well. The changes can be found on page 8, Paragraph 2, line 308. Comments 16: In line 225: Missing full stop at the end of the sentence Response 16: Thank you for pointing it out. We agree with this comment. The full stop was added at the end of the sentence. The changes can be found on page 8, Paragraph 2, line 312 Comments 17: In line 231: A lonely ‘P’ at the end of the line Response 17: Thank you for pointing it out. We agree with this comment. The fragment was corrected. The changes or corrections can be found on page 8, Paragraph 2, line 303 Comments 18: In line 234: Why does the name ‘Xylanase’ start with a capital letter? Response 18: Thank you for pointing it out. We agree with this comment. The xylanase was corrected. The changes or corrections can be found on page 9, Paragraph 1, line 319 Comments 19: In line 237: Not very well thought out sentence Response 19: Thank you for pointing it out. We agree with this comment. The sentence was improved. The changes or corrections can be found on page 9, Paragraph 1, lines 324-225 Comments 20: In line 244, 245: Wrong way to refer to literature Response 20: Thank you for pointing it out. We agree with this comment. The references were corrected and the paragraph was revised. The changes or corrections can be found on page 10, Paragraph 1, line 337 Comments 21: In line 249: The value in brackets should be separated by a space Response 21: Thank you for pointing it out. We agree with this comment. The brackets were added and the numbers were separated. The changes or corrections can be found on page 10, Paragraph 2, lines 341-343. Comments 22: In line 251: No spaces in the fragment: ‘…2.05mg.g-1 …’ Response 22: Thank you for pointing it out. We agree with this comment. The space was added and well-revised. The changes or corrections can be found on page 10, Paragraph 2, line 343. Comments 23: In line 252: ‘Miah et el’ should be ‘Miah et al.’ Response 23: Thank you for pointing it out. We agree with this comment. The citation of reference was revised to ‘Miah et al.’ The changes or corrections can be found on page 10, Paragraph 2, lines 344. Comments 24: In line 259: Too many spaces in the fragment: ‘…reported by Dedousi et al. …’; No spaces in the fragment: ‘…Dedousi et al.(2024)…’ Response 24: Thank you for pointing it out. We agree with this comment. The citation of reference was revised to ‘Dedousi et al.(2024). The corrections can be found on page 10, Paragraph 3, line 351. Comments 25: Please standardize the form of units. The author uses the ‘U/g’ notation once and ‘mg.g-1‘ the next time. Response 25: Thank you for pointing it out. We agree with this comment. Unit forms were standardized according to the samples measured. The mg. g-1 was used for the nutrient composition in the fruiting bodies and the Unit per gram (U/g) was used in the enzyme activity. The change can be found on page 6, 7, lines 235,274 Comments 26: Charts on Figures 1-4 have fonts that are too large compared to the font in the text. In the captions of figures 1-4 we can again find that Pleurotus ostroatus is P969, etc. It is unnecessary. Figure 1 caption: Why does the name ‘Laccase’ start with a capital letter? In the figure you can see an increase in enzyme activity. Why wasn't the experiment extended to assess its maximum? Response 26: Thank you for pointing them out. We agree with the comments. The font of the figures was revised according to the text. The stain codes were described in full as figure legend; laccase was revised in writing by lower letter, substrate depletion in growth medium over extended periods. The five enzyme activities were analysed and revised carefully. The change can be found on page 5 line 190; Pages 7,8,9, Figure 1,2,3,4,5 Comments 27: The most difficult point to understand is the fragment (line 222) where the author writes that the highest peak activity occurs on the 10th day of cultivation. Analyzing graph 4 we cannot agree with this. Explanation needed. Response 27: Thank you for pointing it out. We agree with this comment. The statistical analysis data was revised. MnP activity was analysed carefully. The correction can be found on page 7 line 314.
|
||

Round 2
Reviewer 1 Report
Comments and Suggestions for Authors
Agradezco a los autores por su amable respuesta a los comentarios realizados y por considerarlos para mejorar la calidad del documento.
Para su publicación se requiere atender algunas observaciones que han sido marcadas en morado dentro del documento revisado y que de manera general se enumeran a continuación:
1. Se recomienda verificar que los nombres científicos indicados tengan el formato correcto: Iniciar el género con mayúscula y la especie con minúscula.
2. En la metodología se recomienda revisar cuidadosamente si los autores utilizaron la técnica ELISA para determinar lacasas, CMCasas y xilanasas o si utilizaron métodos convencionales a microescala (por ejemplo, la oxidación de ABTS en microplacas). En caso de ser así, se determinan las terminaciones de las lacasas durante un minuto.
3. La determinación final de MnP no es precisa, es decir, si utilizaron un kit, se debe incluir la empresa comercial para conocer los reactivos mencionados al buscar la referencia. También se recomienda que se utilice el mismo formato para referirse a los reactivos (indicados en morado).
4. Respecto a la expresión de mL, se debe homogeneizar (indicado en el documento)
5. Separar algunas unidades del valor numérico
6. Homogeneizar la representación del valor p, ya sea P ≤ 0,05 o P≤0,05.
7. Evite utilizar juicios de valor como “máximo”. Se recomienda cambiar por “superior”, “más significativo” o máximo.
8. En las tablas no queda claro el criterio de asignación de las letras, si “a” corresponde a la más alta o a la más baja. Se recomienda revisar y utilizar el mismo criterio para todas las tablas.
9. En las figuras se recomienda dejar espacio entre las leyendas de los ejes y los paréntesis de las unidades.
10. Aún falta aclarar la eficiencia del proceso de pasteurización, es decir, indicar si no hubo contaminaciones en los tratamientos, ya que la presencia de otros organismos puede afectar los niveles, principalmente de lacasa. Por esta razón, se recomienda aclarar si hubo o no presencia de otros organismos que pudieran afectar el resultado.
11. Se recomienda enriquecer la discusión para resaltar la importancia de los resultados obtenidos.

Author Response
Response to Reviewer Comments
Thank you very much for taking the time to review this manuscript and providing useful comments. I have tried to translate the comments into English to facilitate a better understanding of the valuable comments you have provided for enhancing the manuscript. Kindly find the detailed responses below and in the resubmitted revised version.
Comments 1: It is recommended to verify that the scientific names indicated have the correct format: Start the genus with a capital letter and the species with a lowercase letter.
Response 1: All Comments highlighted in the PDF files related to scientific names were revised accordingly. The resubmitted version is uploaded to the system. Thank you for pointing out them. We agree with the comments and were very useful.
Comments 2: In the methodology, it is recommended to carefully review whether the authors used the ELISA technique to determine laccases, CMCase, and xylanases or if they used conventional microscale methods (e.g., ABTS oxidation in microplates). If so, the laccase determinations are made within one minute.
Response 2: Thank you for pointing out them. We agree with the comments. The oxidation of the ABTS was measured by determining the increased absorbance at 420nm which needs to be recorded every 30 seconds for 180 seconds the following formula was used to calculate laccase activity Enzyme Activity Definition: The quantity of enzyme required to oxidize 1 nmol of substrate ABTS per gram of sample per minute is one unit of enzyme activity. Based on this methody to determine laccase we have use used conventional microscale methods. The changes can be found on page 4, paragraph 1 lines 142-148.
Comments 3: The final determination of MnP is not precise; if a kit was used, the commercial company should be included to identify the reagents mentioned when looking for the reference. It is also recommended to use the same format when referring to reagents (indicated in purple).
Response 3: Thank you for pointing out this. We agree with the comments. The final determination of MnP was revised accordingly. The changes can be found on page 4, paragraph 3 lines 157-164.
Comments 4: Regarding the expression of mL, it should be homogenized (indicated in the document) revised line 159.
Response 4: Thank you for pointing them out. We agree with the comments. The unit was homogenized in whole revised versions. Different corrections can be found on page 4, in lines 137-140,159-164,166-167,177,186,188.
Comments 5: Separate some units from the numerical value.
Response 5: Thank you for pointing them out. We agree with the comments. The unit was separated into whole revised versions. The changes can be found in line 177.
Comments 6: Homogenize the representation of the p-value, either P ≤ 0.05 or P ≤ 0.05.
Response 6: Thank you for pointing them out. We agree with the comments. The p-values were revised accordingly to P≤0.05) in the whole revised manuscript. The changes can be found in lines 205-211.
Comments 7: Avoid using value judgments such as "maximum." It is recommended to change it to "superior," "more significant," or "maximum."
Response 7: Thank you for pointing them out. We agree with the comments. The recommendations were taken into consideration, we have used “most significant” in the manuscript. The changes can be found in lines 213,226,227,245.
Comments 8: In the tables, the criterion for assigning letters is not clear, whether "a" corresponds to the highest or the lowest. It is recommended to review and use the same criterion for all tables.
Response 8: Thank you for pointing this out. We agree with the comments. The assigning letters were revised in tables “a” corresponds to the highest value. The changes can be found in lines 217, 234.
Comments 9: In the figures, it is recommended to leave space between the axis legends and the parentheses of the units.
Response 9: Thank you for pointing this out. We agree with the comments. All figures were revised by leaving space in the axis legends and parentheses of the units.
Comments 10: The efficiency of the pasteurization process still needs to be clarified, i.e., indicate if there were no contaminations in the treatments, as the presence of other organisms can affect the levels, mainly of laccase. Therefore, it is recommended to clarify whether there was the presence of other organisms that could affect the result.
Response 10: Thank you for pointing this out. We employed pasteurized Cenchrus fungigraminus . The methodology involved soaking the dried grass in water with the addition of 2% lime for two hours. After soaking, the grass was removed from the limed water and thoroughly pressed to eliminate excess moisture. The resulting pressed grass was then supplemented with 20% wheat bran and 2% lime and then directly inoculated with the spawn of three oyster strains. During the treatment, the contamination rate was not significant at the level of 0.05 that why we have a reasonable BE in all species.
Comments 11: It is recommended to enrich the discussion to highlight the importance of the results obtained."
Response 11: Thank you for pointing this out. We agree with the comments. The discussion section was improved based on the reviewer's recommendations. The changes can be found in lines 335-386.
Thank you very much

Reviewer 3 Report
Comments and Suggestions for Authors
The previous manuscript described using non-sterilized giant grass to cultivate three Pleurotus species, including P. ostreatus, P. florida, and P. pulmonarius, and investigating their growth rate, yields, and enzyme activities. However, the revised manuscript changed the substrate from a non-sterilized giant grass, previously thought to be the Pennisetum sinese, to the pasteurized Cenchrus fungigraminus used to cultivate the mushrooms. It’s inconceivable, which is the main reason for further evaluation. It is not recommended for Current Issues in Molecular Biology. In addition to the main reason mentioned above, some of the others are listed below.
1. The use of giant grass as a cultivating material was not a novel idea and has been studied, including the authors’ previous publication (Ref 34, Niyimbabazi et al. 2022). Indeed, these two works were very similar, though revised. Moreover, the authors replied that the previous author referenced in the comments utilized a composted substrate, which differs from the one employed in this article. However, the materials used in this study were also composted substrates.
2. The enzyme activities for five enzymes were still not statistically analyzed, and the data had been drastically modified compared to the previous version.
3. The results were still not compared with those using other materials. For example, C. fungigraminus was used to evaluate the enzymatic reactions of P. ostreatus in the authors’ previous publication (Ref 21, Claude et al. 2024), but the differences were not mentioned.
4. The citations in the text still didn’t follow the journal’s rule.
Author Response
Response to Reviewer Comments
Thank you very much for taking the time to review this manuscript and providing useful comments. Kindly find the detailed responses below and in the resubmitted revised version.
Comments 1: The previous manuscript described using non-sterilized giant grass to cultivate three Pleurotus species, including P. ostreatus, P. florida, and P. pulmonarius, and investigating their growth rate, yields, and enzyme activities. However, the revised manuscript changed the substrate from a non-sterilized giant grass, previously thought to be the Pennisetum sinese, to the pasteurized Cenchrus fungigraminus used to cultivate the mushrooms. It’s inconceivable, which is the main reason for further evaluation. It is not recommended for Current Issues in Molecular Biology. In addition to the main reason mentioned above, some of the others are listed below.The use of giant grass as a cultivating material was not a novel idea and has been studied, including the authors’ previous publication (Ref 34, Niyimbabazi et al. 2022). Indeed, these two works were very similar, though revised. Moreover, the authors replied that the previous author referenced in the comments utilized a composted substrate, which differs from the one employed in this article. However, the materials used in this study were also composted substrates.
Response 1: Thank you for pointing those comments out.
- In the previous article, you said in the comment, I served as the second author alongside the first author. In that study, we utilized a substrate made from composted giant grass, which underwent a 21-day composting period for microbe disinfection. During this time, we monitored the temperature of the compost heap by turning it every five days and recording the daily temperature. Following the composting period, we allowed the material to cool down for 3days. The final substrate formula consisted of composted giant grasses mixed with 10% wheat bran and 2% gypsum powder, resulting in a moisture content of 58% before inoculating with the spawn of oyster strains.
In contrast, this article. We employed pasteurized Cenchrus fungigraminus, in this research. Our methodology involved by soaking the dried grass in 2% limed water for two hours. After soaking, the grass was removed from the limed water and thoroughly pressed to eliminate excess moisture. The resulting pressed grass was then supplemented with 20% wheat bran and 2% lime directly inoculating with the spawn of three oyster strains.
- Giant Juncao grass, the scientific name is Cenchrus fungigraminus, was invented by Professor Lin Zhanxi, who is a co-author in this manuscript and the name was accepted in 2022 in the Journal of Forest and Environment, 2022, listed in the International Plant Name Index. Link: https://www.ipni.org/n/urn:lsid:ipni.org:names:77320785-1.This means the grass we have used in this manuscript is not Pennisetum sinese.
- To use this grass (Cenchrus fungigraminus) is novel because is a perennial grass with high cellulose and crude protein content, capable of yielding up to 200 tons per hectare as biomass, it has abilities to grow at a wide range of temperatures utilizing various lignocelluloses which is very necessary for mushroom cultivation and it also helps reduce the use of wood in mushroom cultivation which causes deforestation.
- We have decided to adopt a scientific name in this manuscript in response to feedback from reviewers who expressed difficulty in finding the botanical name for giant juncao grass and was also suggested that we utilize a pasteurized method instead of the non-sterilized method we originally employed, as it aligns better with the scientific methodology we have used.
Comments 2: The enzyme activities for five enzymes were still not statistically analyzed, and the data had been drastically modified compared to the previous version.
Response 2: Thank you for pointing out them. We agree with the comments. The enzyme activities for five enzymes were statistically analysed in the revised resubmitted. The changes can be found in the figure 1,2,3,4,5.
Comments 3: The results were still not compared with those using other materials. For example, C. fungigraminus was used to evaluate the enzymatic reactions of P. ostreatus in the authors’ previous publication (Ref 21, Claude et al. 2024), but the differences were not mentioned. Line 269
Response 3: Thank you for pointing out this. We agree with the comments. The results were compared with previous work by Claude et al. (2024). The changes can be found in the discussion section in the line 365-367.
Comments 4: The citations in the text still didn’t follow the journal’s rule.
Response 4: Thank you for pointing out this. We agree with the comments. In the resubmitted manuscript the citations in the text and reference followed the journal’s rules.
Thank you very much.
